

# An Automated Baseline Correction Protocol for Infrared Spectra of Atmospheric Aerosols Collected on Polytetrafluoroethylene (Teflon) Filters

Adele Kuzmiakova[1], Ann M. Dillner[2], and Satoshi Takahama[1]

[1]ENAC/IIE Swiss Federal Institute of Technology Lausanne (EPFL), Lausanne, Switzerland
[2]University of California, Davis, Davis, California, USA

*Correspondence to:* S. Takahama (satoshi.takahama@epfl.ch)

**Abstract.** A growing body of research on statistical applications for characterization of atmospheric aerosol Fourier Transform Infrared (FT-IR) samples collected on polytetrafluoroethylene (PTFE) filters (e.g., Russell et al., 2011; Ruthenburg et al., 2014) and a rising interest in analyzing FT-IR samples collected by air quality monitoring networks call for an automated PTFE baseline correction solution. The existing polynomial technique (Takahama et al., 2013) is not scalable to a project

with large amount of aerosol samples because it contains many parameters and requires expert intervention. Therefore, the question of how to develop an automated method for baseline correcting hundreds to thousands of ambient aerosol spectra given the variability in both environmental mixture composition and PTFE baselines remains. This study approaches the question by detailing the statistical protocol, which allows for the precise definition of analyte and background sub-regions, applies non-parametric smoothing splines to reproduce sample-specific PTFE variations, and integrates performance metrics

from atmospheric aerosol and blank samples alike in the smoothing parameter selection. Referencing 794 atmospheric aerosol samples from seven Interagency Monitoring of PROtected Visual Environment (IMPROVE) sites collected during 2011, we start by identifying key FT-IR signal characteristics, such as non-negative absorbance or analyte segment transformation, to capture sample-specific transitions between background and analyte. While referring to qualitative properties of PTFE background, the goal of smoothing splines interpolation is to learn the baseline structure in the background region to predict

the baseline structure in the analyte region. We then validate the model by comparing smoothing splines baseline corrected spectra with uncorrected and polynomial baseline (PB) corrected equivalents via three statistical applications: 1) clustering analysis, 2) functional group quantification, and 3) thermal optical reflectance (TOR) organic carbon (OC) and elemental carbon (EC) predictions. The discrepancy rate for a four-cluster solution is 10%. For all functional groups but carboxylic COH the discrepancy is $\leq 10\%$. Performance metrics obtained from TOR OC and EC predictions ($R^2 \geq 0.94\%$, bias $\leq 0.01$

$\mu g\, m^{-3}$, and error $\leq 0.04\ \mu g\, m^{-3}$) are on a par with those obtained from uncorrected and PB corrected spectra. The proposed protocol leads to visually and analytically similar estimates as those generated by the polynomial method. More importantly, the automated solution allows us and future users to evaluate its analytical reproducibility while minimizing reducible user bias. We anticipate the protocol will enable FT-IR researchers and data analysts to quickly and reliably analyze a large amount of data and connect them to a variety of available statistical learning methods to be applied to analyte absorbances isolated in

atmospheric aerosol samples.



# 1 Introduction

Measurement and quantification of atmospheric aerosol composition and abundance provide a basis from which we can monitor regional air quality, predict potential impacts on health and climate, and deduce formation-mechanisms to reduce uncertainties in climate models for simulating alternative scenarios relevant to climate change adaptation or policy decision-making (Drouet et al., 2015; Monks et al., 2009; Isaksen et al., 2009; Goldstein and Galbally, 2007; Kanakidou et al., 2005). Atmospheric aerosols, or particulate matter (PM), occur as complex mixtures of inorganic salts, crustal elements, sea spray, organic compounds, black carbon, and water (Seinfeld and Pandis, 2006), and a combination of analytical techniques are required to resolve their physical and chemical characteristics (Kulkarni et al., 2011). A useful and relative inexpensive strategy is to collect atmospheric aerosol particles onto a substrate for offline analysis in the laboratory. Amongst different substrates, polytetrafluoroethylene (PTFE) filters have been extensively used in both measurement campaigns (Maria et al., 2002, 2003; Takahama et al., 2011; Frossard et al., 2014; Russell, 2003) and routine monitoring networks, such as the IMPROVE network in pristine and rural areas or the Chemical Speciation Network/Speciation Trends Network in urban and suburban areas in the US (Dillner and Takahama, 2015a). Advantages of PTFE substrates include their stability, hydrophobicity, and negligible carbon gas adsorption (Turpin et al., 1994; Gilardoni et al., 2007; Ruthenburg et al., 2014). As such, they are amenable to gravimetric mass, elemental analysis, and detailed chemical speciation analysis (e.g., Surratt et al., 2007).

Carbonaceous particulate matter (PM) composition collected on PTFE filters is characterized by Fourier Transform Infrared (FT-IR) spectroscopy. Organic functional groups in PM absorb mid-infrared (IR) radiation in specific segments of the spectrum. The amount of light absorbed is proportional to the moles of the functional group (Beer-Lambert law). The absorption at the characteristic frequency of a particular type of bond is measured directly through PTFE filter (Maria et al., 2003; Griffiths and De Haseth, 2007). The high frequency region ($>1500$ cm$^{-1}$) contains stretching and bending modes of important functional groups, such as alkane (consisting of saturated aliphatic C-CH bonds found in hydrocarbon chains), carboxylic acid (COH and C=O found in carboxylic acids and diacids), carbonyl (C=O found in ketones, aldehydes, and esters), hydroxyl (COH found in straight chain alcohols), and amine (C-NH$_2$ found in primary amines) (Russell et al., 2011). The fingerprint region ($< 1500$ cm$^{-1}$) contains absorption bands organonitrate (CONO$_2$) and organosulfate compounds (COSO$_3$) (Day et al., 2010; Hawkins and Russell, 2010) but is outside the scope of our study.

A growing number of papers in recent years has been published to introduce and apply different statistical applications for atmospheric aerosol characterization from the infrared spectra. One of the applications include unsupervised clustering of discrete spectra categories to quantify source contributions, such as fossil fuels, biomass vegetation, or biomass burning, to the total organic PM mass. Spectral clustering has been used in several atmospheric aerosol measurement campaigns and data analysis studies (Russell et al., 2009; Liu et al., 2009; Takahama et al., 2011; Ruthenburg et al., 2014). Cluster analysis of spectra have compared favorably with source class interpretation from factor analysis (Russell et al., 2009; Liu et al., 2009; Takahama et al., 2011; Russell et al., 2011), which attribute variations in the spectra matrix to varying contributions from an underlying set of components, and multiple linear regression with predetermined factor sources (Takahama et al., 2011). Another approach, which has a long record in use for quantification of functional group composition and source apportionment



in atmospheric aerosol samples (e.g., Russell et al., 2011), is fitting individual Gaussian lineshapes to quantify alcohol COH, carboxylic COH, alkane CH, carbonyl CO, and amine NH functional groups (Takahama et al., 2013). Finally, functional groups (Coury and Dillner, 2008; Ruthenburg et al., 2014) and carbonaceous aerosol content equivalent to that of thermal optical reflectance (TOR) (Dillner and Takahama, 2015a, b) have been estimated from partial least squares (PLS) calibration

applied to infrared spectra. However, all applications but PLS regression require baseline corrected infrared spectra (without PTFE interferences) to apply the Beer-Lambert law-type analysis and account for variations in analyte (aerosol) absorbance only. Aside from these statistical applications, removing PTFE interferences is a necessary step for visual inspection and comparison of similarity of aerosol composition in FT-IR spectra.

The problem of background removal is ubiquitous in nearly all spectroscopies (e.g. FT-IR, Nuclear Magnetic Resonance

(NMR) and Raman spectroscopies) and their respective applications that quantify chemical quantities based on the shape and distribution of spectral peaks (Schulze et al., 2005; Rinnan et al., 2009; Bacsik et al., 2004) A general formulation of the problem is to partition an observed spectroscopic signal into two components: one that varies smoothly (baseline) and one that is zero except in specific, localized regions (analyte). Yet, background correction represents an ill-posed problem; we do not know the exact proportions of the baseline and analyte in the observed signal. As a result, a realistic approach is to implement

a baseline model representation capable of capturing underlying physical phenomena causing the baseline specific to the spectroscopy type. While many such investigations have been made in FT-IR biospectroscopy (Baker et al., 2014; Trevisan et al., 2012; Felten et al., 2015), single-compound, gas-phase FT-IR (Shao and Griffiths, 2007; Griffiths et al., 2009; Zhao et al., 2015), NMR (Golotvin and Williams, 2000; Xi and Rocke, 2008), and Raman spectroscopies (Weakley et al., 2012; Liland et al., 2010; Rowlands and Elliott, 2011a), the background removal in FT-IR atmospheric aerosol samples remains a far

less-studied topic. Therefore, we evaluate existing classes of background correction methods to identify the most promising one based on ambient aerosol spectral characteristics.

Existing techniques include frequency decomposition (via Fourier transform, wavelets, or digital filters) to separate the baseline component from the analyte absorption in the frequency domain (Shao et al., 2003). While the frequency decomposition techniques have been shown to successfully correct biological (Trevisan et al., 2012) or single-compound FT-IR spectra (Shao

and Griffiths, 2007; Griffiths et al., 2009), they do not apply in the PM context, where spectral features are not well separated due to broad analyte absorption regions in condensed-phase aerosol samples. Another existing class, numerical differentiation (e.g. first or second differentiation, or Savitzky-Golay derivation) (Schulze et al., 2005; Rinnan et al., 2009) leads to noise amplification and requires additional smoothing that is sensitive to the signal to noise ratio for a specific set of samples. Furthermore, as a result of negative values from the derivative transformation, transformed spectra are difficult to visually interpret

for spectroscopists. The interpolation approach (Liland et al., 2010; Ruckstuhl et al., 2001; Mazet et al., 2005; de Rooi and Eilers, 2012; Schirm and Watzig, 1998; Peng et al., 2010; Takahama et al., 2013) uses sample-specific PTFE signal on background regions where analyte absorption is not expected, and interpolates through analyte regions to identify their relative contributions at each wavelength.

Widely-used interpolation methods for aerosol analysis on PTFE filters (Maria et al., 2003; Gilardoni et al., 2007; Takahama

et al., 2013) are not scalable to projects with a large amount (hundreds of thousands) of aerosol samples. The most modern





implementation of these methods (Takahama et al., 2013) addresses the challenges described above by prescribing a set of four default background regions and polynomial model for the variation. Background regions can be adjusted for each sample to improve accuracy; how- ever, this leads to additional costs in labor and variability across users. Alternatively, for a fixed background region, a non-negativity constraint can be imposed to alleviate issues of unrealistic spectral features that can arise

from incorrect specification of the background. Yet, this adaptation to handle negative analyte absorbance can lead to positive bias in certain regions of ambient sample spectra, or overall in blank sample spectra for which the mean absorbance should be zero (as an average of positive and negative values). Finally, as the predefined polynomial forms are unable to account for all PTFE interferences, the method requires subtraction of blank filter spectra to remove some PTFE features a priori Therefore, the baseline correction is performed on the residual spectra rather than the original. However, because blank filters themselves

exhibit variability, there is no perfect PTFE reference and the subtraction procedure may impart additional bias. Additionally, collecting blank PTFE filters increases FT-IR analysis costs and time.

However, a separation of atmospheric aerosol absorbance bands from the PTFE baseline via interpolation can be very complex, and therefore difficult to quantify precisely and reason about. We break down the issue into two separate problems. The first problem is determining sample-specific bounds for analyte and background sub-regions in the atmospheric aerosol

spectra. Atmospheric PM mixtures are thought to comprise $10^4 - 10^5$ atmospheric organic species (Hamilton et al., 2004; Goldstein and Galbally, 2007; Kroll et al., 2011), leading to broad, overlapping IR absorption bands (features on the order of $10 - 10^2$ cm$^{-1}$) of different functional groups that absorb within similar wavenumber regions (Coury and Dillner, 2008). In Figure 1 we show 794 atmospheric PM samples collected on PTFE filter, each differentiated by color. The overlapping absorbance bands can be seen as smoothly-varying features in regions at ∼3700–2200, and ∼1820–1500 cm$^{-1}$, superimposed

on a sloping baseline. The range is only indicative; the wavenumber specificity is further limited by a variability in ambient PM mixture composition. As composition varies as a function of PM source and date, several of these functional groups may be absent in the sample at hand. Due to the absence of structurally distinguishable features to indicate the onset of analyte contributions, it is challenging to pinpoint the exact locations of analyte absorption. The second problem is reproducing the structure of the PTFE baseline. PTFE scattering represents the largest source of variation of the FT-IR signal when particles

are collected (McClenny et al., 1985). The extent of variation in slope and shape of baseline can vary substantially among individual samples (Figure 1). Baseline variations due to PTFE fiber stretching are unique to each sample and do not follow a prescribed or universal pattern, rendering standardized baseline pre-processing methods, for example prescan subtraction, standard normal variate, multiplicative scattering correction (Rinnan et al., 2009), insufficient. Due to a lack of structural specificity of the underlying PTFE signal, we need a sample-adaptive model.

Naturally, this problem raises the question of how to develop an automated method for baseline correcting hundreds or thousands ambient aerosol FT-IR spectra given the variability in environmental mixture composition and PTFE baselines. This study approaches the question by detailing the statistical protocol, which allows for the precise definition of analyte and background sub-regions, applies non-parametric smoothing splines to model sample-specific PTFE variations, and integrates performance metrics from PM and blank samples alike in the smoothing parameter selection. Referencing an extensive set

of atmospheric aerosol samples, in Section 2 we start by identifying key FT-IR signal characteristics (such as non-negative





absorbance or analyte segment transformation), which reduce signal variations to fundamental features to capture sample-specific transitions between background and analyte. To reproduce sample-specific variations in PTFE background and analyte structures, we develop a nonparametric, adaptive model: interpolation based on smoothing splines regulated by the roughness parameter. While referring to qualitative properties of the baseline (such as smoothness), the goal is to learn the baseline

structure in the background region to predict the baseline structure in the analyte region. In Section 3 we evaluate the model both at the physical and application layers. We establish the initial model feasibility by using near-zero blank absorbance and non-negative analyte absorbance as our physical criteria. Further, by comparing smoothing splines baseline (SSB) corrected spectra with polynomial baseline (PB) corrected spectra via three different applications, 1) visual and clustering analysis, 2) functional group quantification, and 3) organic and elemental carbon prediction, we are able to discern which variations in

quantities obtained from SSB corrected spectra are due to inherent variations already present and which are added due to the new baseline approximation. We close with a summary of the baseline correction procedure extendible to the fingerprint region or spectra acquired on other substrates in Section 4.

## 2   Methods

Section 2.1 introduces smoothing splines in the context of FT-IR signal. Section 2.2 and 2.3 detail the modeling protocol,

including formalizing bounds for analyte and background regions and selecting smoothing parameters. Sections 2.4 and 2.5 describe the dataset and applications we used for smoothing splines model evaluation.

### 2.1   Smoothing Splines Model Description

The proposed interpolation method uses smoothing splines, a popular non-parametric regression technique, which has been applied in different steps in spectral signal analysis: data exploration, model building, testing parametric models, and diagnosis

(Rouh et al., 1993; Rowlands and Elliott, 2011b; Poullet et al., 2007; Persson et al., 1992; Katajamaa and Oresic, 2007; Fourmond et al., 2009). Their expression is obtained by minimizing the following two-part objective function

$$\underset{\hat{\boldsymbol{y}}}{\text{minimize}} \sum_{j=1}^{n} w_j(y_j - \hat{y}_j)^2 + \lambda \int_{a}^{b} (\hat{y}''(x))^2 dx \, , \tag{1}$$

where $w_j$ is weight at wavenumber $j$, $y_j$ and $\hat{y}_j$ are observed and fitted absorbances at wavenumber $j$, and $\lambda$ is a smoothing penalty. Minimizing this criterion over the entire spectrum leads to a unique solution, which is a natural cubic spline with knots

at the unique values of the $x_j, j = 1, 2, \ldots, N$ (Hastie et al., 2009). The explicit solution in form of the natural spline eliminates knot-selection problem without leading to over-parameterization due to the smoothing penalty constraint. The advantage of smoothing splines is their capacity to operate both locally, through $w_j$ representations for each wavenumber, and globally, through a single $\lambda$ representation over the entire wavenumber domain.

The first, least squares term, $\sum_{j=1}^{n} w_j(y_j - \hat{y}_j)^2$, represents the similarity measure consisting of the squared distance between

observed absorbance values and interpolating function values. The advantage of locally moderated weights lies in allowing us



to choose whether absorbance at a particular wavenumber $j$ should be included in determining the fitted baseline. We define weights as follows. Let us decompose the original spectral signal into a two-component mixture:

$$y_j = \begin{cases} B_j + A_j & \text{if } j \in \mathcal{W}_A \text{ (analyte region)} \\ B_j & \text{if } j \in \mathcal{W}_B \text{ (background region) .} \end{cases} \tag{2}$$

Here $\boldsymbol{B_j}$ denotes the background component comprising baseline, noise, and, if present, any remaining local, high-frequency
interference (Takahama et al., 2013), $A_j$ denotes the analyte component, $\mathcal{W}_A$ denotes the set of wavenumbers with analyte absorbance, and $\mathcal{W}_B$ denotes the set of wavenumbers without analyte absorbance. We then wish to select observations that represent solely the background component and exclude those that contain analyte contribution:

$$w_j = \begin{cases} 0 & \text{if } j \in \mathcal{W}_A \\ 1 & \text{if } j \in \mathcal{W}_B \text{ .} \end{cases} \tag{3}$$

Other conceptually analogous variants for determining weights exist. Some researchers define weights as posterior proba-
bilities from mixture models (de Rooi and Eilers, 2012). Some researchers use curve fitting with asymmetric weights (Liland et al., 2011; Felten et al., 2015; Peng et al., 2010; Mazet et al., 2005). While differing in the requirement of a priori knowledge on the assignment of observations to different components, all frameworks, including ours, propose that greater weight is given to those observations representing the background only and smaller or no weight is given to those containing contribution from analyte peaks. Therefore, the aim of the least squares term is to extract the structural information from the neighboring
background regions to infer the baseline structure in the analyte region.

The second term of the objective criterion, $\lambda \int_a^b (\hat{y}''(x))^2 dx$, is a regularization term. It constrains $\hat{y}$ to vary smoothly on a global level. Overall, the objective function trades off fit to the spectral data with the smoothness via the tuning parameter, $\lambda$. For smaller values of $\lambda$ more weight is given to fitting the squared error term of the criterion. When $\lambda = 0$ the unique minimizer is a natural cubic spline, which will interpolate the original response, $y_j$. Conversely, for greater values of $\lambda$ more weight is
given to keeping the curvature small. When $\lambda \to \infty$, the unique minimizer is a $2^{nd}$ degree polynomial. A spectrum of $\lambda$ values ranging from 0 to $\infty$ will generate a family of models, from interpolation to the parametric polynomial model.

When faced with a problem of how much smoothing should be applied to fit the spectral data on hand, effective degrees of freedom (EDF) represents a more physically-interpretable metric to parameterize the regularization of the smoothing spline than $\lambda$ (Cantoni and Hastie, 2002). Consider writing the $n-$vector of fitted values, $\hat{y}$, as:

$$\hat{y} = N(N^T N + \lambda \Omega_N)^{-1} N^T y = S_\lambda y \text{ .} \tag{4}$$

Here $N$ denotes an $n \times n$ design matrix of the cubic spline basis functions evaluated at the observed values $x_j$ and $\Omega_N$ is $\int_l^m N_l''(x) N_m''(x) dx$. A linear operator referred to as a smoother matrix, $S_\lambda$ differentially shrinks influence of $y$ toward their



alignment with the corresponding basis functions. Consequently, the EDF of a smoothing spline are defined as the sum of eigenvalues of $\boldsymbol{S}_\lambda$:

$$EDF_\lambda = \sum_{j=1}^{n} \{\boldsymbol{S}_\lambda\}_{jj} \ . \tag{5}$$

EDF is bounded between 2 and $n$. If $\lambda = 0$, $\boldsymbol{S}_\lambda$ becomes the $n \times n$ identity matrix, and $EDF_\lambda = n$. Conversely, if $\lambda = \infty$,
5   $\boldsymbol{S}_\lambda$ becomes the projection matrix from linear regression on $\boldsymbol{x}$, and $EDF_\lambda = 2$. The advantage of reformulating the smoothing parameter in EDF over $\lambda$ is that its span is bounded and defined with respect to the number of wavenumbers in the region we want to baseline.

If a desired (target) EDF is defined by a user, smoothing splines models are usually fitted via the backfitting algorithm to search for the actual EDF closest to the target. At convergence, the solution can be formulated as:

$$EDF_A = \underset{\lambda}{\mathrm{argmin}} \left( EDF_T - \sum_{j=1}^{n} \{\boldsymbol{S}_\lambda\}_{jj} \right)^2$$

where $EDF_A$ represents the actual EDF determined from $\sum_{j=1}^{n} \{\boldsymbol{S}_\lambda\}_{jj}$ (Eq 5) which minimizes the departure from the target EDF, $EDF_T$. The backfitting procedure is implemented in the smooth.spline function of the R statistical package (R Core Team, 2014), which we used to develop our baseline correction model.

Summarizing in Table 1, we argue that smoothing splines offers more adaptive and realistic basis for modeling PTFE vari-
ations than the current method by combing local and global representations. We apply smoothing splines to specific segments where each analyte region is sandwiched by neighboring background regions containing smoothly-varying baseline. As a result, each segment then contains an accurate basis for baseline prediction in the analyte region using an optimal smoothing parameter, $EDF^*$.

### 2.2   FT-IR Baseline Correction Protocol

Using smoothing splines theory described above, we formalize the baseline correction protocol in Table 2. The weights $w_j$ from Eq 3, i.e. $w_j = 0$ in the analyte region $\mathcal{W}_A$ and $w_j = 1$ in the background region $\mathcal{W}_B$, are determined by sample-specific bounds for analyte and background regions, $W_1$ to $W_4$. Figure 2 illustrates a roadmap for our protocol. In **Step 1**, we divide a raw spectrum into 2 segments. Segment 1 includes domain from 4000 to 1820 cm$^{-1}$, to capture the maximum extent of the background regions surrounding the first analyte region. Segment 2 includes domain from 2000 to 1500 cm$^{-1}$ and captures a
sufficient extent of background regions surrounding the second analyte region. Extensions of the upper background bound can made up to 2220 cm$^{-1}$ (to minimize the interference from CO$_2$ band not associated with PM composition (Pavia et al., 2008)) but may require higher $EDF^*$ to capture the background variability.

In **Step 2**, we perform a geometric transformation, which will be used to determine and verify some of the bounds for analyte and background regions: $W_1$ in Segment 1 and $W_3$ to $W_4$ in Segment 2. As a linear operation, this geometric transformation





preserves the actual absorbance magnitudes. Let $\boldsymbol{a}$ denote an $N$-vector of raw absorbances corresponding to a segment selected in Step 1 illustrated in Figure 2. First we rotate $a_j$ about a point $a_1$ such that $a_1 = a_1^R = a_N^R$, where $a_j^R$ denotes the rotated vector element and $\boldsymbol{R}$ denotes the corresponding rotation matrix:

$$\boldsymbol{a}^R = \boldsymbol{R}\boldsymbol{a}, \text{ where } \boldsymbol{R} = \begin{bmatrix} cos(\theta) & sin(\theta) \\ -sin(\theta) & cos(\theta) \end{bmatrix} \text{ and } \theta = \arctan(\frac{\nu_N - \nu_1}{a_N - a_1}) \, .$$

Second, we translate $a_j^R$ such that $a_1^* = a_N^* = 0$, where $a_j^*$ denotes the resulting translated vector:

$$\boldsymbol{a}^* = \boldsymbol{a}^R - a_1^R \, .$$

Projecting raw absorbances on the local platform axis ($a_1 = a_N = 0$) offers a valuable means of numerically representing a raw spectrum, without appealing to underlying PTFE structural specification. The geometric transformation is a key component in our protocol. First, it allows us to analytically separate background from the analyte in $W_4$ by determining a local minimum.

Second, it provides visually-recognizable verification valuable for further method developments, if need be (e.g. precise $W_1$, $W_3$, and $W_4$ are be difficult to recognize in raw data in Figure 1). For instance, the concept is extendible to application developments for baseline correction in the fingerprint region (Day et al., 2010), which is outside the scope of our current study.

In **Step 3**, we determine specific bounds, $W_1$ to $W_4$, for analyte and background regions, $\mathcal{W}_A$ and $\mathcal{W}_B$. The benefits of

15 determining sample-specific $W_1$ and $W_4$ are twofold. First, certain analytes may be absent from a complex aerosol mixture at hand, thereby increasing $\mathcal{W}_B$. Second, higher loadings may lead to broader tails of certain absorption profiles, thereby decreasing $\mathcal{W}_B$. Section 2.3.1 details a method to determine these bounds.

In **Step 4**, we subtract final baselines from transformed segments and stitch the baseline corrected segments together. In the overlapping region between 2000 and 1820 cm$^{-1}$, we use the mean absorbance in the final result. The absorbance between the

20 rightmost background region down to 1500 cm$^{-1}$ is set to zero.

## 2.3 Selection of Model Parameters

The problem of selecting model parameters, $W_1 - W_4$ and EDF, carries key implications for the quality of fitted baselines. Our goal is to select model parameters to reproduce the structure of sample-specific PTFE variations while minimizing physically unrealistic FT-IR features, such as negative absorbance from PM spectra or absorbance from blank spectra. Referencing

an extensive set of baseline corrected ambient and blank samples (described in Section 2.4), we identify two common physical expectations, to which generated baseline should conform: 1) non-negative analyte absorbance and 2) near-zero blank absorbance.

### 2.3.1 Determining Bounds for Analyte and Background Regions

We determine $W_1$ iteratively for each value of the smoothing parameter to satisfy a non-negativity constraint near the bound-

30 aries. Initial (conservative) estimate of $W_1 = 3720$ cm$^{-1}$ is congruent with our understanding of the absence of absorption





bands over the subdomain between 4000 and 3720 cm$^{-1}$(Pavia et al., 2008). Yet, smaller contributions from certain functional groups, such as alcohol OH, increases the likelihood of negative background absorbance if $W_1$ remains underspecified. Therefore, we begin with the initial estimate (grey baseline in Figure 2 Step 3) and iteratively decrease $W_1$ until the non-negativity constraint is satisfied or until $W_1$ reaches $W_2$. We set $W_2$ to 2220 cm$^{-1}$, which universally marks the start of $CO_2$ absorbance band (Pavia et al., 2008). Similarly, we set $W_3$ to 1820 cm$^{-1}$, which universally marks the start of carbonyl absorbance band observed in all PM samples.

To accommodate the specifications of individual samples, $W_4$ is determined as $\tilde{\nu}$ for which $a_j^*$ attains its minimum over the set of candidate values between 1520 and 1600 cm$^{-1}$:

$$W_4 = \operatorname*{argmin}_{j} \left\{ a_j^* : \tilde{\nu}_j \in [1520, 1600] \right\} \tag{6}$$

where $a_j^*$ are transformed absorbances from Step 2. To minimize the interference from the neighboring alkane peak, starting to absorb around 1510 cm$^{-1}$(Pavia et al., 2008), we limit the lower background region to a single wavenumber adjacent to $W_4$, $W_4 - 1(\Delta\tilde{\nu})$.

### 2.3.2 Selection of EDF

To parameterize the influence of EDF on the quality of fitted baselines via the two expectations, we derive two EDF-optimizing metrics: 1) negative absorbance fraction for ambient samples and 2) total normalized absolute blank absorbance for blank filters. We summarize the metrics in Table 3.

Negative absorbance fraction (NAF) represents the contribution of negative analyte absorbance, $\|\boldsymbol{a}_{A-}\|_1$, to the total analyte absorbance, $\|\boldsymbol{a}_A\|_1$:

$$NAF = \frac{\|\boldsymbol{a}_{A-}\|_1}{\|\boldsymbol{a}_A\|_1} \times 100\%$$

where $\|\cdot\|_1$ denotes the 1-norm magnitude of a vector (summation of all absolute values of vector elements). NAF is calculated across the entire wavenumber range in the analyte part of in a given segment, excluding the $CO_2$ absorbance band.

Total normalized absolute blank absorbance, $\|\boldsymbol{a}_B\|_1^*$, quantifies the model's departure from the true result, zero absorbance, per wavelength in a given segment. It is calculated as a 1-norm magnitude of blank absorbances, $\|\boldsymbol{a}_B\|_1$, normalized by the number of wavenumbers in the corresponding wavenumber range (Table 3), $n_{\tilde{\nu}}$:

$$\|\boldsymbol{a}_B\|_1^* = \frac{\|\boldsymbol{a}_B\|_1}{n_{\tilde{\nu}}} .$$

$\|\boldsymbol{a}_B\|_1^*$ is calculated across the entire wavenumber range in a particular segment excluding the $CO_2$ absorbance band. We select $EDF^*$ by evaluating minima from both $\|\boldsymbol{a}_B\|_1^*$ and NAF. Figures 3 and 4 in Section 3.1 present qualitative and quantitative evaluation of resulting spectra for varying $EDF_T$.





## 2.4 Experimental Data

We apply smoothing splines baseline correction to 794 particulate matter ($\leq 2.5\mu$g in diameter, PM$_{2.5}$) samples collected on PTFE filters and 54 blank PTFE filters. The particulate matter samples were collected at IMPROVE sites on every third day in 2011. IMPROVE absorption spectra had been used in a previous studies (Ruthenburg et al., 2014; Dillner and Takahama, 2015a,

b), which detail the mechanics of FT-IR spectra collection. More important for this study is the level of spectral preparation applied prior to the background correction. Following the practice established in (Dillner and Takahama, 2015a, b) we use unmodified spectra in which values interpolated during the zero-filling process were removed. Prior to applying the smoothing splines baseline, we truncate the original wavenumber domain between 4000 and 420 cm$^{-1}$to capture the subdomain between 4000 and 1500 cm$^{-1}$(1,944 wavenumbers). As a reference, the same subdomain is used in the polynomial method (Takahama

et al., 2013). In contrast to Takahama et al. (2013), we do not apply smoothing to remove water vapor interference and carbon dioxide to minimize the number of preprocessing steps.

## 2.5 Applications for model evaluation

**Cluster analysis.** Each spectrum is normalized by its 2-norm magnitude and grouped according to the hierarchical clustering algorithm of Ward (1963). There are inherent differences in the vapor artifacts between the PB corrected and SSB corrected

spectra that are not critical for the algorithms used for quantification of functional groups, or TOR organic and elemental carbon but influence clusters formed from the naïve clustering approach described above. As the PB corrected signal requires differencing the IR spectrum of the PTFE before and after sample collection, water vapor and CO$_2$ signals remaining in the PF corrected spectra represent differences in concentrations present in the chamber during both scans, whereas SSB corrected spectra contain only the amount present in the latter. Therefore, regions where these artifacts are present ($\tilde{\nu} > 3600$ cm$^{-1}$and

$\tilde{\nu} < 2400$ cm$^{-1}$in Segment 1) are excluded from the normalization and clustering, though some water vapor artifact overlapping with analyte absorption remains in Segment 2. In addition, seven samples with specific features or low signal-to-noise ratios are removed from the set prior to the clustering as they are not well discriminated by the algorithm, or influences the grouping of the rest of the spectra.

**Peak-fitting.** We evaluate peak areas fitted to SSB corrected spectra according to parameter constraints described by (Taka-

hama et al., 2013). Peak areas correspond to integrated absorbances from lineshapes fitted for alcohol COH, carboxylic COH, alkane CH, carbonyl CO, and amine NH. We examine the comparability and implications of replacing the PB correction approach with SSB correction in future analyses of this type.

**Prediction of TOR organic carbon (OC) and elemental carbon (EC).** We compare TOR OC and EC predicted from SSB corrected spectra with those predicted from raw and PB corrected spectra. TOR OC and EC are predicted by the PLS method.

PLS regression had been used for prediction on IMPROVE 2011 and IMPROVE 2013 samples in previous studies (Dillner and Takahama, 2015a, b; Reggente et al., submitted), which detail the mechanics of PLSR calibration. To assess the quality of predictions, we use identical metrics established in these studies (Dillner and Takahama, 2015a, b): bias, error, normalized error, coefficient of variation, minimum detection limit (MDL), % below MDL, precision, and blank mean. Likewise, we use





identical procedure to build calibration and test sets (Dillner and Takahama, 2015a, b) from smoothing SSB spectra: calibration and test sets contain two thirds and one third, respectively, of 794 IMPROVE 2011 samples (chronologically stratified within each site).

## 3 Results

At the physical level, we evaluate feasibility of our model by selecting the optimal smoothing parameters in Section 3.1 and by presenting the sample-specific bounds for analyte and background regions in Section 3.2. At the application level, we begin our evaluation of smoothing splines baseline corrected spectra with visual and cluster analysis in Section 3.3, followed by functional group quantification analysis in Section 3.4, and predicted TOR OC and EC analysis in Section 3.5.

### 3.1 EDF selection

Qualitatively, in Figure 3 both PM and blank samples show increasing sensitivity to the amount of smoothing applied. With increasing $EDF_T$, baseline corrected ambient spectra begin to exhibit negative analyte absorbance (left column). Simultaneously, baseline corrected blanks in the region at $3700 - 2500$ and $1820 - 1600$ cm$^{-1}$ begin to depart from our target, zero absorbance (right column).

Quantitatively, in Figure 4 we evaluate the impact of $EDF_T$ on negative absorbance fraction $NAF$ and total normalized
absolute blank absorbance $\|\boldsymbol{a_B}\|_1^*$, introduced in Section 2.3.2. In segment 1 we find that $EDF_T$ between 2 and 4 minimize $NAF$ and its variance simultaneously (median $NAF < 0.01\%$, $97^{th}$ percentile = 0.44%). With respect to blank absorbance, we find $2 \leq NAF \leq 4$ generates reasonably low $\|\boldsymbol{a_B}\|_1^*$ (mean = $3.42 \times 10^{-4}$, $3\sigma = 2.79 \times 10^{-4}$); differing by < 1.5% from the lowest $\|\boldsymbol{a_B}\|_1^*$ corresponding to $EDF_T = 5$. Of the two metrics, we prefer to minimize NAF over $\|\boldsymbol{a_B}\|_1^*$ as NAF represents a more robust metric (the sample size of NAF is an order of magnitude greater and in future applications the choice of smoothing
parameter will likely affect disproportionally more PM samples than blank samples). To finalize the choice of $EDF_T$ from $2 \leq NAF \leq 4$, we now consider the distributions of $EDF_A$ for these target values in Figure 5 A and B.

The extensive number of knots ($x_j$ for which $w_j \neq 0$ from Eq 1) to form bases for fitting splines create limitations on minimum achievable EDF when $EDF_T$ is low (< 7 in segment 1 and < 3 in segment 2). For instance, if we apply baselines with $EDF_T = 4$ in segment 1 (Figure 5 A and B), $EDF_A$ will span between 4.9 and 6.1 depending on the number of basis-
forming knots (Figure S2). However, applying baselines with target < 4 will lead to identical solutions, confirming that the set of $EDF_A$ is indeed the minimum achievable EDF in the search domain. Therefore, out of $EDF_T$ candidates for $EDF^*$ we choose 4 as it represents the true underlying parameters most accurately ([4.9, 6.1] for PM samples and [4.9, 4.9] for blank samples).

In segment 2 (Figure 4 B) we start by limiting the evaluation to $EDF_T$ values where the $97^{th}$ percentile of $NAF < 0.22\%$
(roughly a half of the value from the best-fit model in segment 1): $4 \leq EDF_T \leq 7$. Out of this subset, we find selecting 4 as $EDF_T$ minimizes the second metric, $\|\boldsymbol{a_B}\|_1^*$ (mean = $1.71 \times 10^{-4}$, $3\sigma = 1.06 \times 10^{-4}$) in Figure 4 D. Also importantly, 4



represents the most parsimonious solutions without visually distorting the blank baseline and shape of the PM peaks (Figure 3). By selecting $EDF^*$ 4, now the actual EDF parameters match the desired parameter (Figure 5 C and D).

## 3.2 $W_1$ and $W_4$

Figure 6 presents empirical cumulative distributions functions of $W_1$ and $W_4$ from PM and blank samples. Distribution of $W_1$ in PM samples spans values between 3300 and 3710 cm$^{-1}$ with 50 % of samples having $W_1 > 3700$ cm$^{-1}$, reflecting on sample-specific PM mixture composition (illustration of spectra in Figure 3 A). $W_1$ in blank samples was determined to be 3710 cm$^{-1}$ (Figure 3 B). Distribution of $W_4$ in PM samples spans values between 1520 and 1600 cm$^{-1}$, reflecting on sample-specific ammonium absorbance width (Figure 3 A). $W_4$ in blank samples was determined to be 1600 cm$^{-1}$, which is consistent with our physical expectation about zero amine absorbance (Figure 3 B).

## 3.3 Cluster analysis

The number of samples from SSB corrected spectra not sharing the same relative labeling as those from PB corrected spectra vary with the total number of clusters used to partition the spectra set. Figure 7 shows that the discrepancy for 787 samples increases as the set is partitioned into a larger number of clusters. The difference in sample labeling varies between 5% for two clusters and 11% for five clusters; the increase is observed for larger number of clusters because spectra are grouped according to finer variations in their features. Feature (wavenumber) selection and advanced algorithms can lead to more robust clustering that is less sensitive to small variations in spectra (Hastie et al., 2009), but visual comparisons of spectra in the present form of aggregation can provide useful interpretations as discussed below.

Figure 8 shows spectra from the two baseline correction algorithms grouped into categories using the approach described in Section 2. Type I spectra are selected manually, and Types II–V are determined by a four-cluster solution by hierarchical clustering (with discrepancy rate between PB and SSB of 10%). "Type I" spectra display low absorbance in the alcohol COH region, visible methylene paired peaks (2920 and 2850 cm$^{-1}$) from CH$_2$ bonds present in vegetative detritus (Hawkins and Russell, 2010), and the largest absorbance in the carbonyl CO region (centered near 1700 cm$^{-1}$) compared to the rest of the sample spectra. This spectra type indicates a dominant contribution from biomass burning aerosol spectra (Hawkins and Russell, 2010; Takahama et al., 2011). These two samples were collected in St. Marks, FL, during January and February; fire burning is prescribed near this location during January through May of each year. "Type II" spectra also contain sharp methylene peaks but also stronger absorption above 3100 cm$^{-1}$ associated with alcohol COH and less pronounced carbonyl CO absorption. Sixty percent of the 132 SSB corrected spectra are found in Phoenix, AZ, so this is interpreted to be associated with urban aerosol (we note that Phoenix samples may be over-represented in this spectra set as two sampling sites out of the seven analyzed in this work are located in this city). Similar features have been found in spectra from the urban environment of Mexico City (Liu et al., 2009). "Type V" contains spectra for which peaks near 3200–3100 cm$^{-1}$ are most prominent, indicating the significant presence of ammonium. These features have commonly been reported in fossil fuel burning samples or factor analysis components (Hawkins and Russell, 2010; Takahama et al., 2011; Guzman-Morales et al., 2014) that have been assigned by correlation with combustion tracers (e.g., V, Cr, Ni, Zn, As) and backtrajectory analyses. These aerosols presumably arise



from a combination of aged background aerosol and aerosols produced locally in the presence of high oxidant concentrations of polluted environments (Liu et al., 2011). However, 87% of the 322 SSB corrected Type V samples are found in the five non-urban sites, suggesting that in this data set this spectroscopic signature is more indicative of aged secondary aerosol. Ammonium concentrations are often temporally correlated with oxidized organic aerosol (e.g., Jimenez et al., 2009; Lanz

et al., 2010) which increases in abundance toward rural areas (Zhang et al., 2007). Types III and IV share some combination of features from Types I, II, and V, with the ammonium peak near 3200 cm$^{-1}$more visible in Type IV and larger contributions from methylene peaks visible in Type III. The peak near 3700 cm$^{-1}$present in several Type IV spectra is suggestive of phenolic compounds also present in biogenic aerosol (Bahadur et al., 2010).

This analysis demonstrates that the new SSB correction method can generate spectra similar in profile to PB corrected

spectra used in past studies, providing a basis for further mixture analysis.

### 3.4  Peak fitting analysis

Figure 9 presents integrated absorbances for alcohol COH, carboxylic COH, alkane CH, carbonyl CO, and amine NH quantified from PB and SSB corrected spectra. For all functional groups but carboxylic COH the discrepancy between the two methods is < 10% (slope of the regressed line < 1 ±0.1). The difference is on the same order of magnitude as the cluster discrepancy

rate. The bias in carboxylic COH fitting is likely due to the fact that its lineshape was fixed specifically to the PB corrected spectra (Takahama et al., 2013), and is more sensitive to the absorption profile to which it is fitted than the Gaussian peaks with adjustable parameters used for fitting other functional groups. The bias in may be alleviated by rederiving the carboxylic COH lineshape for the smoothing splines method, or applying an adjusted molar absorption coefficient. The bias of 13% is on the order of variation in absorption coefficients of carboxylic COH estimated for different organic acid compounds, and also

within uncertainty for an absorption coefficient estimated from the mean of these values (Takahama et al., 2013).

### 3.5  Prediction of TOR organic and elemental carbon

Figure 10 presents performance metrics from TOR OC and TOR EC predictions obtained from SSB corrected spectra. All fits are characterized by high coefficients of variations ($R^2 \geq 0.94$) and near-zero bias ($\leq 0.01$ $\mu g\,m^{-3}$), demonstrating accurate predictions. With respect to predicted TOR OC, performance metrics from the test set (Figure 10 B) are on a par with those

obtained from raw spectra and PB corrected spectra. Specifically, error (0.09 $\mu g\,m^{-3}$) and normalized error (10%) are on the same order as those obtained from raw spectra (error = 0.08 $\mu g\,m^{-3}$, normalized error = 11%) and PB corrected spectra (error = 0.08 $\mu g\,m^{-3}$, normalized error = 12%; (Dillner and Takahama, 2015a)). In Table 4 we show applying SSB leads to lower MDL (0.06 $\mu g\,m^{-3}$) that is statistically significantly different from the no baseline case (0.14 $\mu g\,m^{-3}$), leaving no samples below MDL. Precision (0.06 $\mu g\,m^{-3}$) obtained from SSB corrected spectra is on the same order as those obtained from raw

(0.12 $\mu g\,m^{-3}$) or PB corrected spectra (0.21 $\mu g\,m^{-3}$).

Likewise, TOR EC performance metrics from the test set (Figure 10 D) are on a par with those obtained from raw spectra and PB corrected spectra. Specifically, error (0.04 $\mu g\,m^{-3}$) and normalized error (27%) are on the same order as those obtained from raw spectra (error = 0.02 $\mu g\,m^{-3}$, normalized error = 21%) and PB corrected spectra (error = 0.04 $\mu g\,m^{-3}$, normalized



error = 24%) (Dillner and Takahama, 2015b). Table 4 shows that MDL (0.01 $\mu g\, m^{-3}$) obtained from SSB corrected spectra is similar to MDL obtained from raw or PB corrected spectra (all $\leq 0.02\ \mu g\, m^{-3}$).

In summary, SSB corrected spectra OC and EC predictions from blank and ambient samples as accurate and precise as those from raw or PB corrected spectra. No additional bias is introduced as a result of SSB correction implementation. However, the reduction in complexity of baseline correction is amenable for scaling up to a large number of samples. To some extent, PLS is a robust regression method and is able to effectively remove contributions to the signal which are not related to the target analyte. While individual predictions vary, we show in Figure S3 the quality of TOR OC and EC predictions are not statistically affected by the choice of EDF between 2 and 30.

## 4   Conclusions

Within the past few years the guided polynomial baseline corrected algorithm has been applied to characterize the ambient FT-IR spectra by classifying mixtures (Russell et al., 2009; Liu et al., 2009; Takahama et al., 2011; Ruthenburg et al., 2014), quantifying organic functional groups (Takahama et al., 2013), and predicting TOR OC and EC (Dillner and Takahama, 2015a, b). Here our results demonstrate, similar estimates (cluster discrepancy rate = 10%, functional group difference $\leq 13\%$, and $R^2 \geq 0.94\%$, bias $\leq 0.01\ \mu g\, m^{-3}$, error $\leq 0.04\ \mu g\, m^{-3}$ in TOR OC and EC predictions) can be obtained using a new, automated baseline correction protocol. In this paper we detailed the statistical framework, which allows for the precise definition of analyte and background sub-regions, applies non-parametric smoothing splines to model sample-specific PTFE variations, and integrates performance metrics from PM and blank samples alike in smoothing parameter selection. The proposed protocol unifies and simplifies many of the steps in existing technique while eliminating the need for expert intervention. More importantly, the automated solution allows us and future users to evaluate its analytical reproducibility while minimizing reducible bias due to current default background regions or a variability in human judgement in adjusting these regions. The solution was developed as a direct response to the growing body of research on statistical applications for characterization of FT-IR atmospheric aerosol samples collected on PTFE filters and a rising interest in analyzing FT-IR samples collected by air quality monitoring networks. As a result, we anticipate the model will enable FT-IR researchers and data analysts to quickly and reliably analyze a large amount of data and connect them to a variety of available statistical analyses outlined in this paper.

Important avenues for future research include extending this approach to the remaining part of mid-IR absorbance spectrum ($1500 - 420$ cm$^{-1}$). The fingerprint region contains important functional groups (Day et al., 2010), such as organonitrates, which can benefit from an adaptive baseline correction algorithm. As demonstrated in this manuscript, the general framework of 1) segmenting baselining regions of interest such that they contain smoothly-varying (or uniformly-sloping) baseline and 2) using FT-IR physical criteria (such as minimal blank absorbance, non-negative analyte/background absorbance, and no baseline discontinuities) for parameter selection can provide a good starting point for these tasks.

*Acknowledgements.*  The authors acknowledge EPFL discretionary funding and funding from IMPROVE program and EPA (National Park Service cooperative agreement P11AC91045). We thank Matteo Reggente and Andrew Weakley for helpful conversations.



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





# Tables

**Table 1.** Comparison of key background modeling characteristics pertaining to the proposed and current models

| Characteristics | Proposed Method | Current Method |
|---|---|---|
| Funtional Form | Smoothing Splines | Polynomial |
| Type | Non-parametric | Parametric |
| Representations | Global ($EDF_T$) and local ($w_j$) | Global ($n^{th}$ degree of a polynomial) |
| Requires prescans? | No | Yes |
| Requires user's input? | No | For every scan |

**Table 2.** The relationship between FT-IR spectrum features and smoothing splines parameters to model those features

| | | Spectrum Characteristics | | Model Parameters | |
|---|---|---|---|---|---|
| Segment | Region Type | Wavenumber Range ($cm^{-1}$) | Type of modeled baseline | Weights | EDF |
| | Background upper | $[4000, W_1]$ | Fitted | $w_j = 1$ | |
| 1 | Analyte | $[W_1, W_2]$ | Predicted | $w_j = 0$ | $EDF^*$ |
| | Background lower | $[W_2, 1820]$ | Fitted | $w_j = 1$ | |
| | Background upper | $[2000, W_3]$ | Fitted | $w_j = 1$ | |
| 2 | Analyte | $[W_3, W_4]$ | Predicted | $w_j = 0$ | $EDF^*$ |
| | Background lower | $[W_4, W_4 - 1(\Delta\tilde{\nu})]^a$ | Fitted | $w_j = 1$ | |

[a] The lower background regions consists of a single wavenumber adjacent to $W_4$ (Section 2.2.1).



**Table 3.** Relationship between fitted baseline characteristics as a result of varying EDF and EDF-optimizing metrics to represent these characteristics

| Segment | | Physical criterion | Sample type | Wavenumber range (cm$^{-1}$) | Representation |
|---------|---|--------------------|-------------|------------------------------|----------------|
| 1 | 1 | Near-zero blank absorbance | Blank | [4000, 2500], [2200, $W_2$] | Total normalized absolute blank absorbance, $\|a_B\|_1^*$ |
|   | 2 | Non-negative analyte absorbance | Ambient | [$W_1$, 2500] | Negative absorbance fraction, $NAF$ |
| 2 | 1 | Near-zero absorbance | Blank | [2000, 1500] | Total normalized absolute blank absorbance, $\|a_B\|_1^*$ |
|   | 2 | Non-negative analyte absorbance | Ambient | [$W_3$, $W_4$] | Negative absorbance fraction, $NAF$ |

**Table 4.** MDL and precision for FT-IR OC and TOR OC.

| Carbon type | Metric | TOR | FT-IR raw spectra[d] | FT-IR PB corrected spectra[d] | FT-IR SSB corrected spectra |
|-------------|--------|-----|----------------------|-------------------------------|------------------------------|
| OC | MDL ($\mu g\, m^{-3}$)[b,c] | 0.05 | 0.14, [0.11, 0.28] | 0.11, [0.08, 0.17] | 0.06, [0.04, 0.09] |
|    | % below MDL | 1.5 | 2.6 | 0.7 | 0.0 |
|    | Precision ($\mu g\, m^{-3}$)[b] | 0.14 | 0.12 | 0.21 | 0.06 |
|    | Mean Blank ($\mu g$) | NR[e] | $0.1 \pm 1.5$ | $1.9 \pm 1.2$ | $0.1 \pm 0.6$ |
| EC | MDL ($\mu g\, m^{-3}$)[b,c] | 0.01 | 0.02, [0.01, 0.02] | 0.01, [0.00, 0.01] | 0.01, [0.01, 0.02] |
|    | % below MDL | 3 | 1 | 2 | 1 |
|    | Precision ($\mu g\, m^{-3}$)[b] | 0.11 | 0.04 | 0.06 | 0.06 |
|    | Mean Blank ($\mu g$) | NR[e] | $0.06 \pm 0.17$ | $0.08 \pm 0.15$ | $0.01 \pm 0.12$ |

[b] Concentration units of $\mu g\, m^{-3}$ for MDL and precision are based on the IMPROVE volume of 32.8 m$^3$. [c] Numbers inside the interval denote 95 % confidence intervals on the estimate. [d] (Dillner and Takahama, 2015a, b). [e] Not reported.

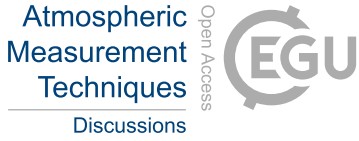

# Figures




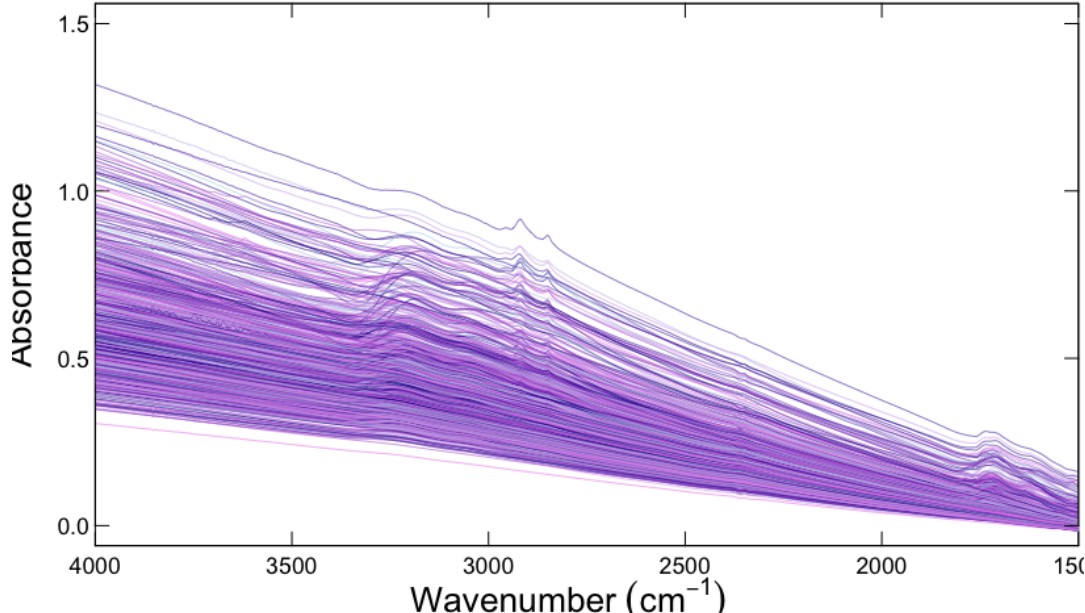

**Figure 1.** 794 FT-IR atmospheric aerosol spectra collected on PTFE filter. Each spectrum is color-differentiated.



**STEP 1:** Divide a raw spectrum into segments.

**STEP 2:** Perform geometric transformations on each segment.

**STEP 3:** Determine bounds for the background and analyte regions in each segment. Select EDF* to apply baselines.

**STEP 4:** Subtract final baselines from transformed spectra and stitch the corrected segments together.

**Figure 2.** 1) Uncorrected spectrum partitioned into two segments. Segment 1: 4000 - 1820 cm$^{-1}$, segment 2: 2000 - 1500 cm$^{-1}$, 2) Transformed segments with zero first and last absorbance values, 3) Upper: initial baseline (grey), final baseline estimated iteratively via a non-negativity constraint (red), red vertical lines delineate background and analyte regions: $W_1 = 3360$ cm$^{-1}$ and $W_2 = 2220$ cm$^{-1}$, lower: final baseline (blue), blue vertical lines delineate background and analyte regions: $W_3 = 1820$ cm$^{-1}$ and $W_4 = 1520$ cm$^{-1}$, 4) Resultant corrected spectrum.

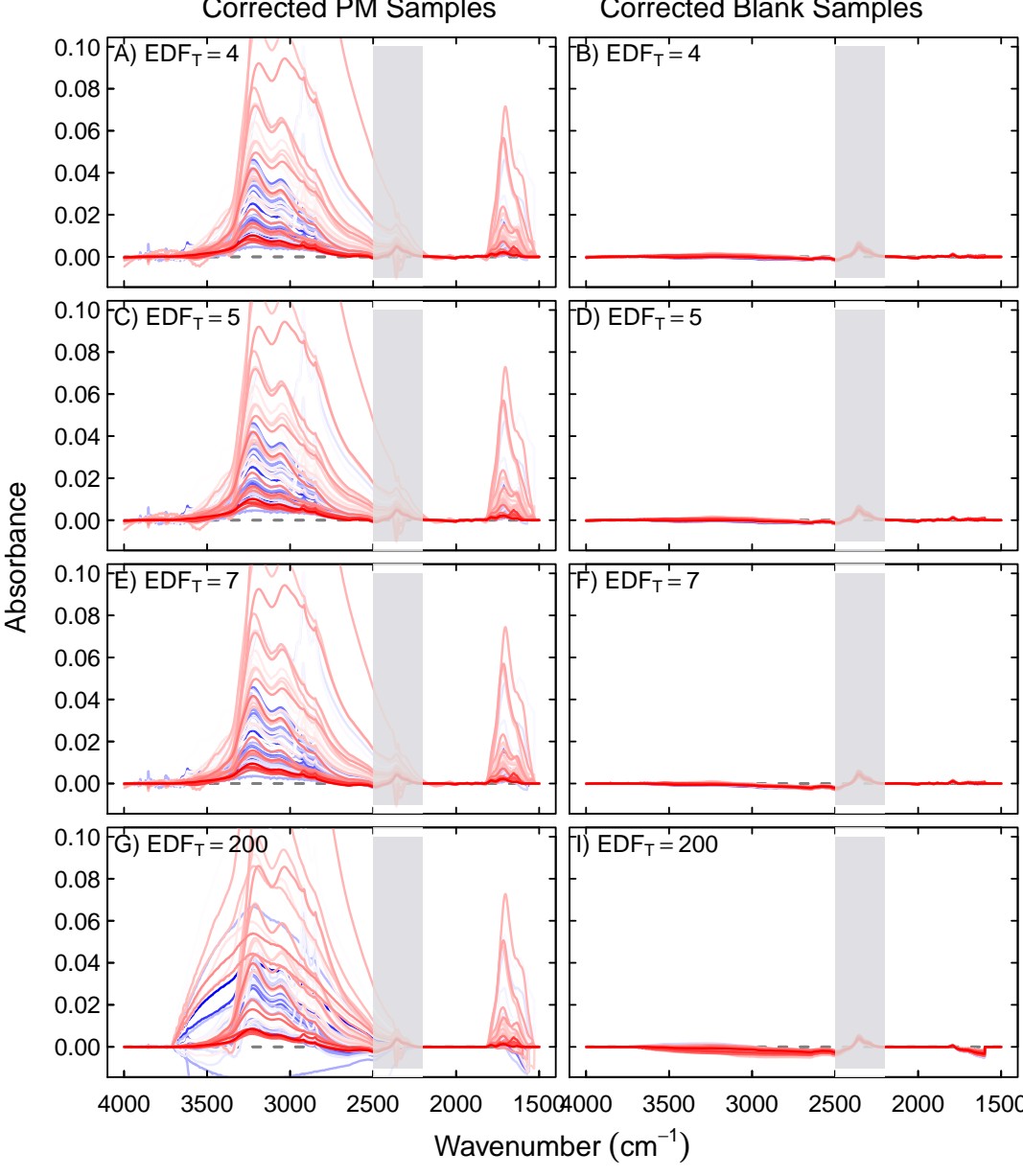

**Figure 3.** 54 randomly selected ambient samples (left) and 54 blank samples (right) corrected by varying $EDF_T$. Each spectrum is color-differentiated. Shaded in color is $CO_2$ absorption band between 2500 and 2220 cm$^{-1}$ not associated with PM composition.





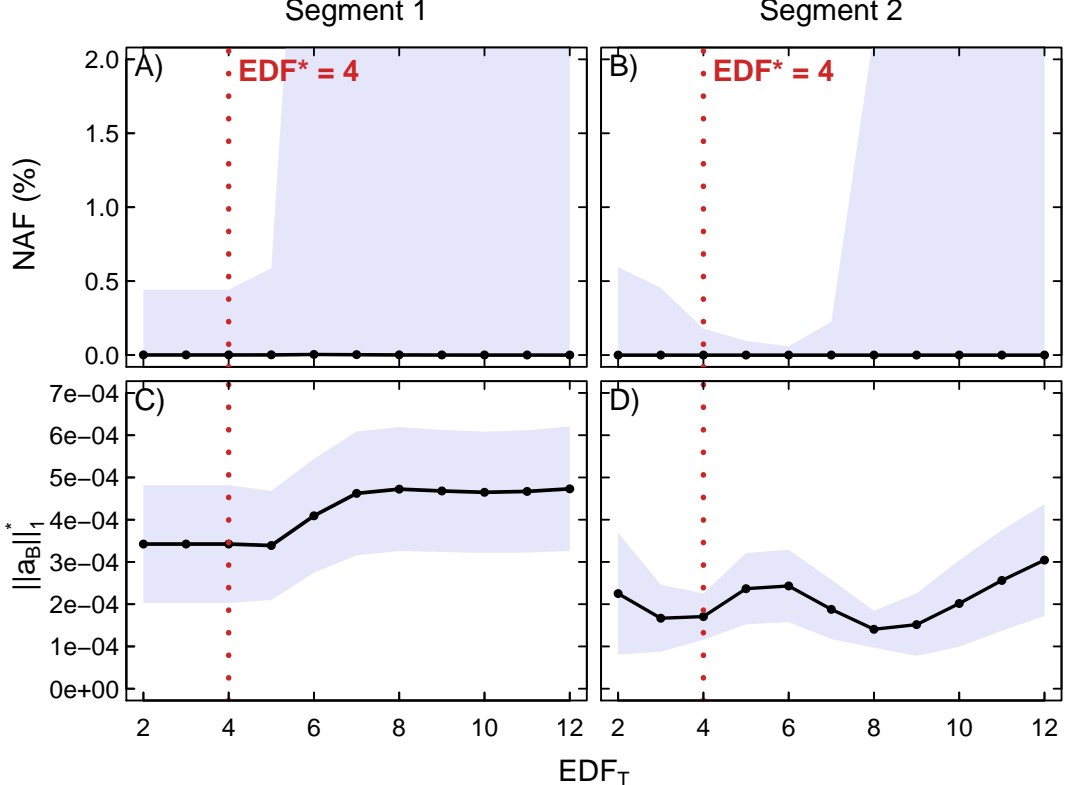

**Figure 4.** Median $NAF$ in segment 1 (A) and segment 2 (B) calculated from 794 ambient samples (black points). Lower and upper bounds of shaded areas denote $3^{rd}$ and $97^{th}$ percentiles. Mean $\|a_B\|_1^*$ for $2 \leq EDF \leq 12$ in segment 1 (C) and segment 2 (D), calculated from 54 IMPROVE 2011 laboratory blank samples (black points). Shaded areas denote three standard deviations from the mean. All: Black line is drawn to capture the overall trend. While we select the interval $2 \leq EDF_T \leq 12$ specifically to highlight each metric's minima, we present results from the entire interval $2 \leq EDF_T \leq n$ for completeness in Figure S1.





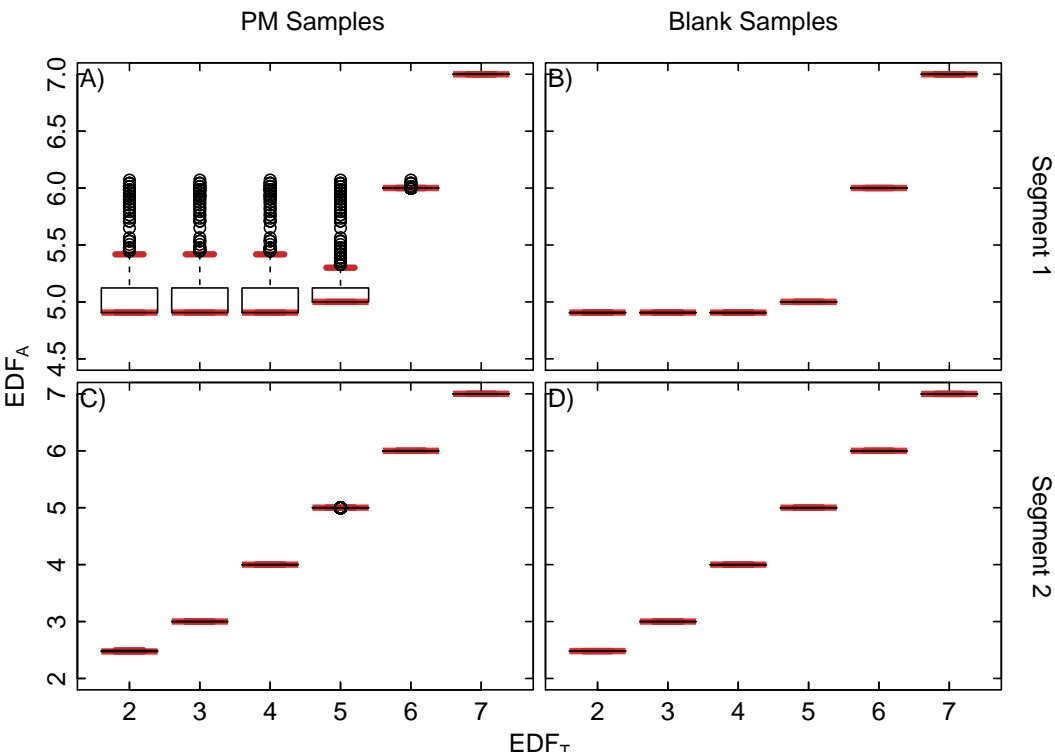

**Figure 5.** Box-and-whisker plots representing distributions of $EDF_A$ for a given $EDF_T$ used in segment 1 (A, B) and segment 2 (C, D) in both PM (n = 794) and blank samples (n = 54). Median and whiskers in each box-and-whisker plots are highlighted in red.





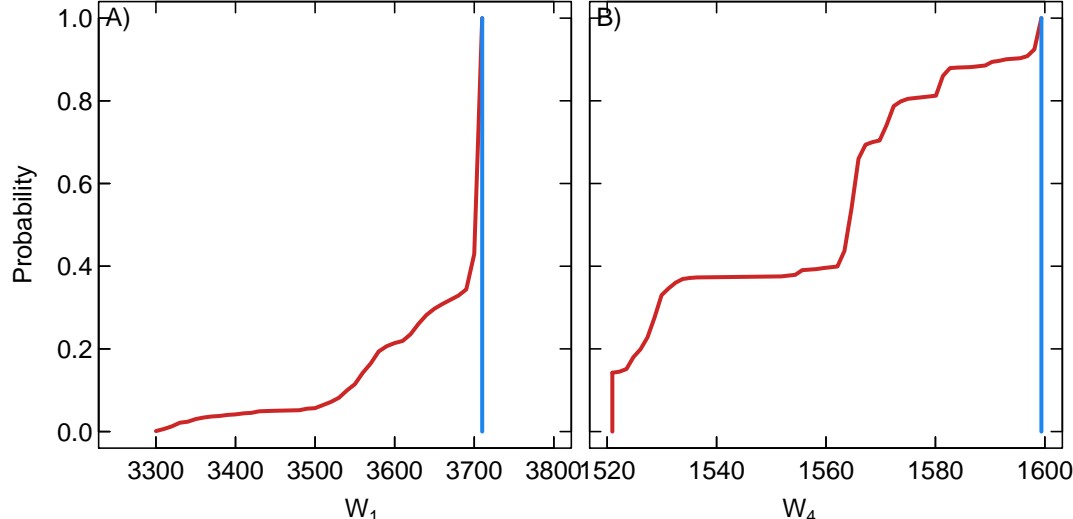

**Figure 6.** Empirical cumulative distribution functions representing distributions of $W_1$ and $W_4$ in PM samples (n = 794) in red and blank samples (n = 54) in blue.





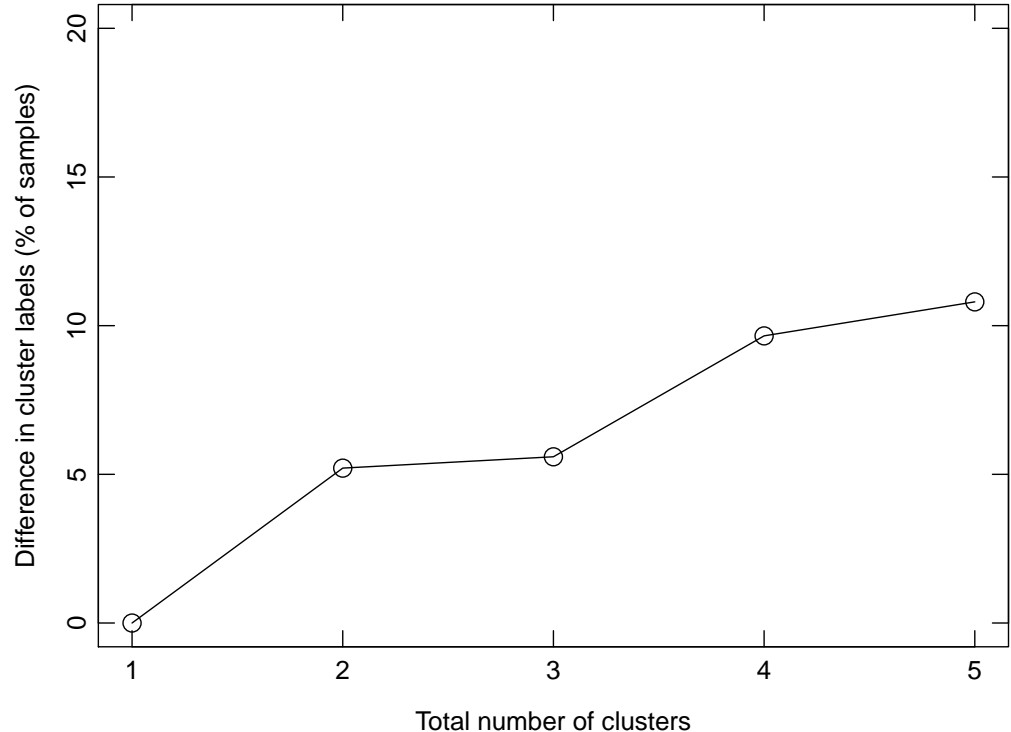

**Figure 7.** The fraction of samples designated into a different cluster when using SSB corrected spectra compared to clusters formed using PB corrected spectra. Total number of samples used in this analysis is $n = 787$.





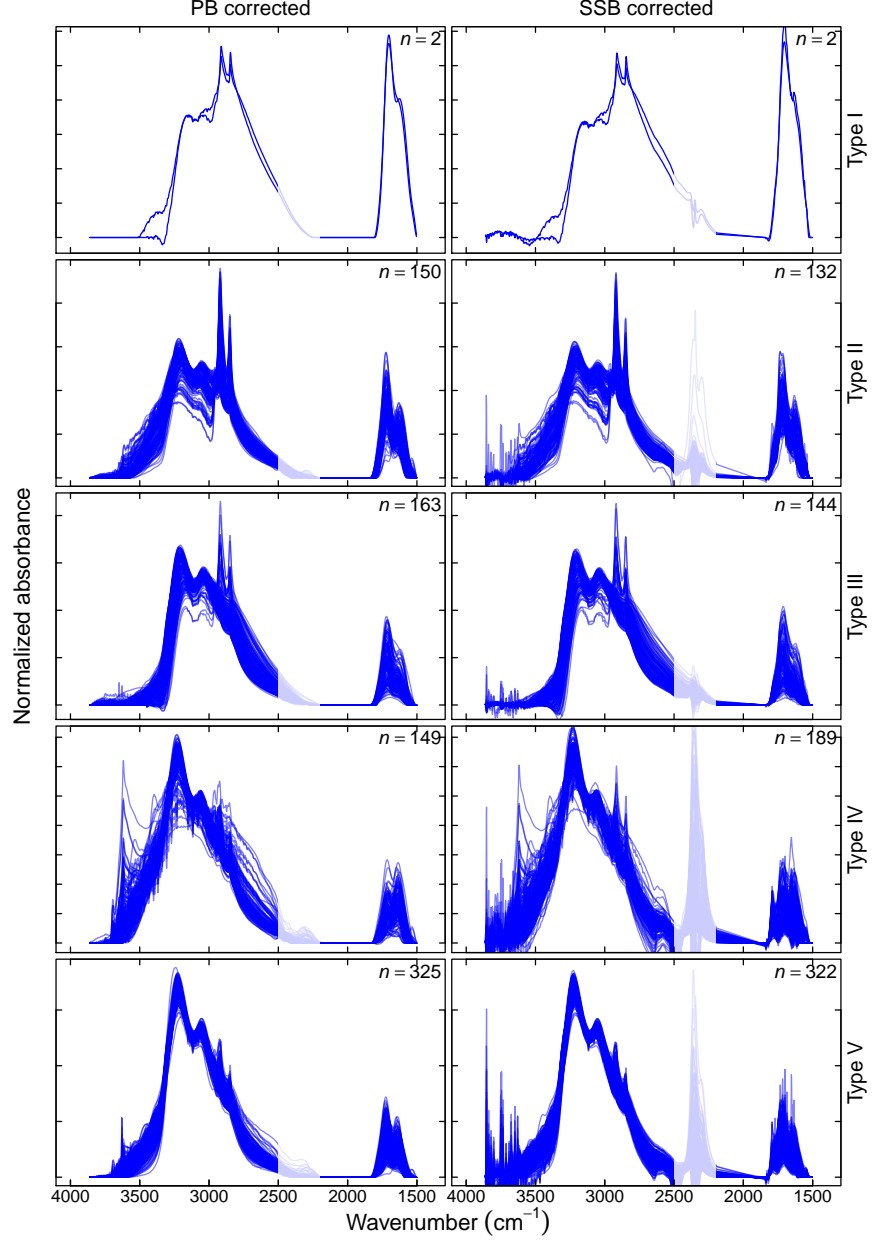

**Figure 8.** Cluster membership for polynomial and smoothing splines methods. The region between 2500–2200 cm$^{-1}$ is masked to indicate region of CO$_2$ absorption not associated with aerosol composition.





**Figure 9.** Integrated peak area corresponding to different functional groups (A - E) from polynomial baseline and smoothing splines baseline corrected spectra. Slope magnitudes represent the slope of the regressed line. Silver line represents a one to one line.





**Figure 10.** Predicted FT-IR OC versus measured TOR OC using smoothing splines corrected spectra for A) calibration set (n=517) and B) test set (n=268). Predicted FT-IR EC versus measured TOR EC using smoothing splines corrected spectra for C) calibration set (n=501) and D) test set (n=268).