# Peer review of "An Automated Baseline Correction Protocol for Infrared Spectra of Atmospheric Aerosols Collected on Polytetrafluoroethylene (Teflon) Filters"

_Atmospheric Measurement Techniques, 2015_

## Referee Comment (RC1) · Anonymous Referee #1 · 24 Feb 2016

Kuzmiakova et al present an automated smoothing splines (SSB) method to perform baseline corrections for PTFE (Teflon) filters analyzed by FT-IR. They apply this method to 794 filter samples collected by the US IMPROVE network. Results using the new SSB method are compared to analysis of the same filters using the manual polynomial baseline correction (PB). The two baseline correction methods yield similar results, and the SSB method has the advantage that it removes user intervention and potential biases associated with human intervention.

The paper is well written and topically relevant to AMT. It will be of most interest to

other researchers who analyze PTFE filters using FT-IR. The method presented in this manuscript may potentially be of interest to all users of FT-IR or other spectroscopic methods. I suggest publication after addressing the comments below.

(1) In Section 2 (Methods), the authors go into great detail to explain the smoothing spline baseline correction. As part of their explanation, they introduce a significant amount of nomenclature. Many of the quantities they introduce seem to have multiple variants (e.g., EDF*, EDF_T, EDF_A, etc). The end result is an extremely detailed description of the method that is nearly impenetrable, as it requires readers to mentally juggle all of the different (and in my case, often unfamiliar) variable names and meanings. I strongly suggest that the authors include a glossary to define all of the various nomenclature, and that they take specific care to differentiate the various flavors of certain quantities (e.g., EDF as noted above).

(2) As a specific example of the previous comment, I am confused by the difference between EDF* and EDF_T. From Section 2.3.2 it seems that EDF* is determined from minimizing NAF (page 9, line 27). However the last line in that section (last line on page 9) introduces EDF_T, and I cannot find an explanation for EDF_T prior to that sentence. Thus, the use of EDF_T in section 3.1 is confusing. On Page 11, Line 21, EDF_A is used, again without prior introduction. While I understand in general terms the goal of using EDF in the fitting, I cannot tell the difference between EDF*, EDF_T, and EDF_A from the text.

(3) The authors are intimately familiar with the analysis of PTFE filters. Unfortunately, they assume that their readers have an equal level of expertise. For example, many readers may not be familiar with OC and EC determinations from PTFE filters, as the referenced papers were only recently published (most readers are likely much more familiar with OC/EC analysis of quartz filters). Thus some additional explanation of OC/EC analysis of PTFE filters is warranted. Likewise, some more background on clustering and its application for PTFE filters would be helpful.

(4) The authors argue that the automated SSB method is preferential because it allows bulk, presumably fast analysis or large sets of filters. Thus it would be useful for the authors to note the time required by the existing PB method (per filter or for the full set of 794 filters) versus the automated SSB method.

(5) Figure 10 shows performance of the calibration set versus the test set, however all other figures seem to indicate the full set of 794 filters. Was a test set only used for OC/EC analysis? I think that a better explanation of OC/EC analysis of PTFE filters (comment 3) could clear up my confusion.

(6) For the data shown in Figures 7-10 - Is the same EDF used for every filter, or is it adjusted automatically for each? If it is the former, do the authors expect that the general EDF guidelines given here (e.g., EDF between 2 and 4) will apply universally? Or will each operator/sample collector need to determine EDF at the outset of analysis?

(7) Page 11, Line 22 - "extensive number of knots" for forming the spline - how exactly are the number and location of knots determined?

(8) Figure 7 may understate the clustering differences between PB and SSB. The total difference for the 5-cluster solution is only 10%, but for certain clusters (e.g., Type IV), the difference is much larger.

(9) Figure 9 compares integrated peak areas for different functional groups after baseline correction with SSB and PB. The figure implies that the same integration method or code is used in each case. The authors should specify whether or not this is true.

(10) Equation 1: what are x and y"?

(11) Page 10, Line 1 - particle size is labeled as micro-gram, not micro-meter

---

## Referee Comment (RC2) · Anonymous Referee #2 · 27 Feb 2016

General Comments This paper presents a method to correct for variability in PTFE filter baseline IR absorbance brought about by non-uniform filters and by the stretching incurred during long sampling periods. This tool will help researchers access data from networks, such as IMPROVE, in order to increase our understanding of organic aerosol transformation. The paper is generally well-written, with only minor grammatical or typographical errors. There are some sections that could use more descriptive, perhaps less mathematical, text in order to reach a wider audience.

I recommend this paper for publication with very minor revisions.

[Figure]

Specific Comments

Beginning in section 2.1, the authors introduce a large number of variables represented by symbols. Given the large number, a list should be provided to help the reader follow along or be able to easily look up any variable in one place.

Section 3.1 is hard to read. Granted, this kind of mathematical treatment is new to me, but the authors might consider using more descriptive text, in addition to the variables, to help a wider range of researchers access this tool. This applies broadly to the paper to some extent, but especially to this section.

I wondered when reading the introduction and conclusion exactly how much time, or computational time, was saved by doing the baseline fitting with spline fits rather than polynomial. It also was not entirely clear how the spline technique alleviated the necessary "expert" interpretation relative to the polynomial fit. If the authors could elaborate on these two points, it would strengthen the paper.

Technical Comments

Abstract line 6: I suggest moving the word "remains" directly after "the question"

Pg 2 line 8: "relatively" Pg 2 line 26: "A growing number of papers have been published in recent years" Pg 2 line 27: "One of the applications includes" Pg 4 line 8: "a priori. Therefore"

Pg 14 line 13: omit comma

Figure 3: x-axis tick mark labels on left and right panels overlap. Figure 7 may be unecessary (especially given the large number of high-information figures that are certain to be large).

---

## Author Comment (AC1) · 23 Apr 2016

**Response to Referee # 1**

April 23, 2016

Kuzmiakova et al present an automated smoothing splines (SSB) method to perform baseline corrections for PTFE (Teflon) filters analyzed by FT-IR. They apply this method to 794 filter samples collected by the US IMPROVE network. Results using the new SSB method are compared to analysis of the same filters using the manual polynomial baseline correction (PB). The two baseline correction methods yield similar results, and the SSB method has the advantage that it removes user intervention and potential biases associated with human intervention.

The paper is well written and topically relevant to AMT. It will be of most interest to other researchers who analyze PTFE filters using FT-IR. The method presented in this manuscript may potentially be of interest to all users of FT-IR or other spectroscopic methods. I suggest publication after addressing the comments below.

*The authors thank the reviewer for the constructive comments and suggestions that help improve the quality of the paper. We have revised the manuscript to address the reviewers comments and made response to each comment in this file. The responses are in red.*

1. In Section 2 (Methods), the authors go into great detail to explain the smoothing spline baseline correction. As part of their explanation, they introduce a significant amount of nomenclature. Many of the quantities they introduce seem to have multiple variants (e.g., $EDF^*$, $EDF_T$, $EDF_A$, etc). The end result is an extremely detailed description of the method that is nearly impenetrable, as it requires readers to mentally juggle all of the different (and in my case, often unfamiliar) variable names and meanings. I strongly suggest that the authors include a glossary to define all of the various nomenclature, and that they take specific care to differentiate the various flavors of certain quantities (e.g., $EDF$ as noted above).

   *We agree that providing an explicit glossary upfront will help with keeping track of variables and their variants introduced throughout Section 2. In this light, we start the Methods section with a summary of notation for most commonly used variables, together with their description. We also break down variables into 3 distinct categories with respect to their roles they play in our smoothing splines model implementation. We include pointers to the sections where the variables are formalized to allow for quick and selective reading.*

   *Regarding the distinction among $EDF^*$, $EDF_T$, and $EDF_A$; $EDF^*$ is chosen from a set of $EDF_T$ by evaluating two metrics: negative analyte absorbance and total normalized absolute blank absorbance, and $EDF_A$ is the actual value of the parameter computed for each $EDF_T$ according to Eq 6. To clarify the confusion, we reformulated the last two sentences in 2.3 which now read:*

   *"We select $EDF^*$ from a range of $EDF_T$ by evaluating minima from both total normalized absolute blank absorbance and $NAF$. To that end, Figures 3 and 4 in Section 3.1 present qualitative and quantitative evaluation for varying $EDF_T$ together with $EDF^*$ selection."*

2. As a specific example of the previous comment, I am confused by the difference between $EDF^*$ and $EDF_T$. From Section 2.3.2 it seems that $EDF^*$ is determined from minimizing $NAF$ (page 9, line 27). However the last line in that section (last line on page 9) introduces $EDF_T$, and I cannot find

an explanation for $EDF_T$ prior to that sentence. Thus, the use of $EDF_T$ in section 3.1 is confusing. On Page 11, Line 21, $EDF_A$ is used, again without prior introduction. While I understand in general terms the goal of using $EDF$ in the fitting, I cannot tell the difference between $EDF^*$, $EDF_T$, and $EDF_A$ from the text.

$EDF_A$ and $EDF_T$ are first introduced on page 7, lines 8-12 in the first version of the manuscript, while the reviewer first notices them on pages 9 and 11. To prevent similar confusion by other readers, we have made reference to Eq 5.5 in the locations on the equivalent of pages 9 and 11 in the revised manuscript.

Essentially, $EDF_T$ is a user-defined smoothing parameter selected before applying the model, eg 2. $EDF_A$ is a model-calculated smoothing parameter that gets as close to $EDF_T$ as possible, e.g. a value of 2.5, depending on the number of data points to be fitted (i.e, $y_j$ for which $w_j$ does not equal to 0, Figure 4). This information is stated on page 7, lines 8-12. Next section on this page, lines 14-18, goes on to introduce $EDF^*$, albeit without providing a clear link to $EDF_T$. To fill the gap, we add a linkage before the last paragraph in Section 2.1, which now reads:

"Thus, the user-defined $EDF_T$ will form a basis for model parameter solutions from which the optimal parameter, $EDF^*$, will be chosen (Section 2.3)."

Also, as stated in our Response to Comment # 1, we modified the closing in Section 2.3 to rephrase that $EDF^*$ will be selected from a set of candidates ($EDF_T$) by studying the optimizing metrics: normalized absolute blank absorbance and $NAF$.

Finally, the opening table in Section 2.1, Table 1, introduces these relationships prior to their formalization in the text. We hope these modifications will lend the paper greater clarity.

3. The authors are intimately familiar with the analysis of PTFE filters. Unfortunately, they assume that their readers have an equal level of expertise. For example, many readers may not be familiar with OC and EC determinations from PTFE filters, as the referenced papers were only recently published (most readers are likely much more familiar with OC/EC analysis of quartz filters). Thus some additional explanation of OC/EC analysis of PTFE filters is warranted. Likewise, some more background on clustering and its application for PTFE filters would be helpful.

We have revised Section 2.5 to include further background:

"Cluster analysis with FT-IR measurements generates natural categories for PM samples based on spectral similarity. These categories can represent mixture classes of chemically complex aerosols, and their association with meteorological and collocated measurements have shown to provide complementary information for source apportionment (Takahama et al., 2011; Corrigan et al., 2013). For this purpose, each spectrum is SSB corrected to isolate the analyte contribution to the IR absorbance, normalized by its 2-norm magnitude to emphasize variation in relative composition rather than absolute concentration, and grouped according to the hierarchical clustering algorithm of Ward (1963)."

"Dillner and Takahama (2015a,b) recently demonstrated that collocated PTFE samples analyzed by FT-IR and quartz fiber filters analyzed by TOR can be used to build calibration models that predict TOR-equivalent OC and EC concentrations from new FT-IR spectra. One of several calibration models with accuracy and precision on a par with TOR precision can be constructed when the concentration range and composition of carbonaceous samples in the calibration set approximately resemble those in the test (challenge) set. For this work, we use an identical procedure as described by Dillner and Takahama (2015a,b) for building calibration and test sets from 794 IMPROVE 2011 samples chronologically stratified within each site. The spectra are SSB corrected and calibration and test samples are drawn to contain two thirds and one third of the entire set, respectively. Only TOR OC and EC predictions necessitate dividing the data set into calibration and test sub-sets; the previous two applications, clustering and peak-fitting, are

4. The authors argue that the automated SSB method is preferential because it allows bulk, presumably fast analysis or large sets of filters. Thus it would be useful for the authors to note the time required by the existing PB method (per filter or for the full set of 794 filters) versus the automated SSB method.

   While this is an important question, we believe time comparison is not entirely applicable to our FT-IR context due to high variability across different users in applying the current polynomial baseline. The polynomial method requires a fair amount of judgement in estimating the baselining intervals in ambient aerosol spectra (e.g. as shown in Figure 1). For example, users with extensive FT-IR baselining experience may feel comfortable using visual inspection to identify the background and analyte regions. Others, on the other hand, may prefer to look at past examples or do a brief literature search on the presence and locations of absorbing functional groups in ambient samples. There is not a clear-cut answer here and the amount of time to apply the polynomial method will likely depend on persons level of expertise and purpose of the analysis. To emphasize this variability, we added an explanation on page 4, lines 3-4:

   "For example, users with extensive FT-IR baseline correction experience may feel comfortable using visual inspection to identify the background and analyte regions in Figure 1. Others, on the other hand, may prefer to look at past examples or conduct a brief literature search on the presence and locations of absorbing functional groups."

   In addition, we have included the following statement in the Conclusion section to address the overall time required:

   "Although the exact reduction in user time may be difficult to generalize due to high variability across different users, we reason the following approximation applies. Qualitatively, if $N$ values are considered for each free parameter in each method, then the amount of time for expert examination of each model solution scales up with $N^4$ for the polynomial method (due to 4 boundary points as free parameters) and $N$ for the smoothing splines method (due to 1 EDF parameter). Also importantly, the evaluation metrics, which we established in this manuscript, have been shown to sufficiently simplify the parameter selection process for users of any level of experience."

5. Figure 10 shows performance of the calibration set versus the test set, however all other figures seem to indicate the full set of 794 filters. Was a test set only used for OC/EC analysis? I think that a better explanation of OC/EC analysis of PTFE filters (comment 3) could clear up my confusion.

   That is correct. The only application requiring splitting the data into calibration and test sub-sets is TOR OC/EC analysis. All other applications were applied to all 794 samples. To clarify this distinction, we added a sentence at the end of Section 2.5 which now reads:

   "Only TOR OC and EC predictions necessitate dividing the data set into calibration and test sub-sets; the previous two applications, clustering and peak-fitting, are applied to the entire data set."

6. For the data shown in Figures 7-10 - Is the same $EDF$ used for every filter, or is it adjusted automatically for each? If it is the former, do the authors expect that the general $EDF$ guidelines given here (e.g., $EDF$ between 2 and 4) will apply universally? Or will each operator/sample collector need to determine $EDF$ at the outset of analysis?

   We thank the reviewer for this relevant question. Yes, in this dataset (IMPROVE data collected in 2011) we used the same $EDF$ for each sample. As Figure 3 demonstrates, the individual differences between $EDF_T$ 4 and 7 in segment 1 are negligible; on the whole these parameters do a very similar job in minimizing the undesirable quantities, such as negative analyte absorbance and blank absorbance, in Figure 4. However, we anticipate we and other FT-IR analysts may benefit

from sample-specific *EDF* when analyzing datasets collected under different conditions, such as different sampling flowrate or filter type. In fact, we are currently working on a paper which will introduce sample-specific *EDF* in the baseline correction to make the method scalable to a variety of sampling conditions without requiring user intervention. To recognize the opportunity in this paper, we modified future outlook in Section 4, which now reads:

"One of the important avenues for future research include implementing sample-specific EDF when the parameter choice affects model performance significantly across samples. As Figure 3 demonstrates, the individual differences between $EDF_T$ 4 and 7 in segment 1 are negligible; on the whole these parameters do a very similar job in minimizing the undesirable quantities, such as negative analyte absorbance and blank absorbance, in Figure 4. However, we anticipate we and other FT-IR analysts may benefit from sample-specific *EDF* when analyzing datasets collected under different conditions, be they different sampling flowrate or filter type. Another line of future work may include extending this approach to the remaining part of mid-IR absorbance spectrum $(1500 - 420 \text{ cm}^{-1})$."

7. Page 11, Line 22 - extensive number of knots for forming the spline - how exactly are the number and location of knots determined?

In this application knots forming bases for fitting splines are wavenumbers in observed absorbances used for fitting splines, that is $x_j$ for which $w_j$ are not 0 in Equation 1. To clarify this, the sentence now reads:

"The extensive number of knots to form bases for fitting splines (that is, wavenumbers in observed absorbances used for fitting: $x_j$ for which $w_j$ are not 0 in Equation 1) create limitations on minimum achievable *EDF*."

8. Figure 7 may understate the clustering differences between PB and SSB. The total difference for the 5-cluster solution is only 10%, but for certain clusters (e.g., Type IV), the difference is much larger.

The reviewer is correct in this statement, but the main point of the comparison is the overall magnitude of discrepancy. The actual differences between individual clusters will also depend on the number of clusters and the type of clustering algorithm. In this evaluation, PB corrected spectra serves only as a reference by convention, and cannot be established to be the correct set of spectra in an absolute sense. To de-emphasize the importance of comparisons for each cluster, have modified the text to read:

"The inter-cluster differences will further depend on the number of clusters and the type of clustering algorithm. Since there is no absolute reference for baseline corrected spectra, these discrepancies speak to the differences between two candidate methods."

9. Figure 9 compares integrated peak areas for different functional groups after baseline correction with SSB and PB. The figure implies that the same integration method or code is used in each case. The authors should specify whether or not this is true.

Yes, that is correct. The same peak-fitting method has been used in both SSB and PB corrected spectra. We clarify this in sentence on page X line Y, which now reads:

"We apply the peak-fitting algorithm based on parameter constraints described by Takahama et al 2013 to both SSB and PB corrected spectra and evaluate the differences between two baseline correction methods by comparing peak areas."

10. Equation 1: what are x and y?

Again, we add Table 1 to list the variables upfront, including x and y. Also, x and y are originally defined in text (page 5 lines 25-27).

11. Page 10, Line 1 - particle size is labeled as micro-gram, not micro-meter

This has been now fixed to micrometer.

**References**

Corrigan, A. L., Russell, L. M., Takahama, S., Äijälä, M., Ehn, M., Junninen, H., Rinne, J., Petäjä, T., Kulmala, M., Vogel, A. L., Hoffmann, T., Ebben, C. J., Geiger, F. M., Chhabra, P., Seinfeld, J. H., Worsnop, D. R., Song, W., Auld, J., and Williams, J.: Biogenic and biomass burning organic aerosol in a boreal forest at Hyytiälä, Finland, during HUMPPA-COPEC 2010, Atmospheric Chemistry and Physics, 13, 12 233–12 256, doi:10.5194/acp-13-12233-2013, 2013.

Dillner, A. M. and Takahama, S.: Predicting ambient aerosol thermal-optical reflectance (TOR) measurements from infrared spectra: organic carbon, Atmospheric Measurement Techniques, 8, 1097–1109, doi: 10.5194/amt-8-1097-2015, 2015a.

Dillner, A. M. and Takahama, S.: Predicting ambient aerosol thermal-optical reflectance measurements from infrared spectra: elemental carbon, Atmospheric Measurement Techniques, 8, 4013–4023, doi:10.5194/amt-8-4013-2015, 2015b.

Takahama, S., Schwartz, R. E., Russell, L. M., Macdonald, A. M., Sharma, S., and Leaitch, W. R.: Organic functional groups in aerosol particles from burning and non-burning forest emissions at a high-elevation mountain site, Atmospheric Chemistry and Physics, 11, 6367–6386, doi:10.5194/acp-11-6367-2011, 2011.

Ward, Jr., J.: Hierarchical grouping to optimize an objective function, Journal of the American statistical association, 58, 236–244, 1963.

---

## Author Comment (AC2) · 23 Apr 2016

**Response to Referee # 2**

April 23, 2016

This paper presents a method to correct for variability in PTFE filter baseline IR absorbance brought about by non-uniform filters and by the stretching incurred during long sampling periods. This tool will help researchers access data from networks, such as IMPROVE, in order to increase our understanding of organic aerosol transformation. The paper is generally well-written, with only minor grammatical or typographical errors. There are some sections that could use more descriptive, perhaps less mathematical, text in order to reach a wider audience. I recommend this paper for publication with very minor revisions.

*The authors thank the reviewer for the constructive comments and suggestions that help improve the quality of the paper. We have revised the manuscript to address the reviewers comments and made response to each comment in this file. The responses are in red.*

1. Beginning in section 2.1, the authors introduce a large number of variables represented by symbols. Given the large number, a list should be provided to help the reader follow along or be able to easily look up any variable in one place.

   *As also suggested by Reviewer 1, we agree that providing an explicit glossary upfront will help with keeping track of variables and their variants introduced throughout Section 2. In this light, we start the Methods section with a summary of notation for most commonly used variables, together with their description. We also break down variables into 3 distinct categories with respect to their roles they play in our smoothing splines model implementation. We include pointers to the sections where the variables are formalized to allow for quick and selective reading.*

2. Section 3.1 is hard to read. Granted, this kind of mathematical treatment is new to me, but the authors might consider using more descriptive text, in addition to the variables, to help a wider range of researchers access this tool. This applies broadly to the paper to some extent, but especially to this section.

   *We agree in some parts of Section 3.1 would benefit from a more qualitative discussion. We have rewritten the section by describing the general result and interpretation from figures. For example, the revised paragraph 1 in Section 3.1 now reads:*

   *"Qualitatively, in Figure 3 we compare the behavior of PM (left panel) and blank sample spectra (right panel) for various values of $EDF_T$ (4, 5, 7, and 200: from top to bottom). In this analysis, we used all 54 blank samples and randomly sampled 54 out of 794 PM samples to keep the counts equal and allow for representative cross-comparison. The trend from top to bottom shows both PM and blank samples exhibit increasing sensitivity to the amount of smoothing applied. With increasing $EDF_T$, baseline corrected ambient spectra begin to exhibit negative analyte absorbance (left column). Simultaneously, baseline corrected blanks in the region at $3700 - 2500$ and $1820 - 1600$ cm$^{-1}$ begin to depart from our target, zero absorbance (right column)."*

   *We also included descriptions of Figure 4 layout to help readers orientate and provide more effective transitions between paragraphs in Section 3.1 and Figure 4. For example,*

   *"Quantitatively, in Figure 4 we evaluate the impact of $EDF_T$ on negative absorbance fraction metric, $NAF$, (top panel) and total normalized absolute blank absorbance metric, $\|\vec{a_B}\|_1^*$, (bottom panel) in segments 1 and 2 (left and right panel). Horizontal panels share the same x-axis and*

vertical panels share the same y-axis to allow for representative cross-comparison. Therefore, each plot in the matrix in Figure 4 corresponds to a unique condition in terms of a metric and segment."

Section 3.1 includes additional pointers and structure modifications interspersed throughout the text. We hope these modifications will lend the paper greater clarity and be understood by a variety of researchers.

3. I wondered when reading the introduction and conclusion exactly how much time, or computational time, was saved by doing the baseline fitting with spline fits rather than polynomial. It also was not entirely clear how the spline technique alleviated the necessary "expert" interpretation relative to the polynomial fit. If the authors could elaborate on these two points, it would strengthen the paper.

This is a significant question. Computationally, the polynomial and spline fitting method are linear operations and are therefore comparable. The reduction in user time is difficult to generalize due to high variability across different users. Qualitatively, we can reason that if N values are considered for each free parameter, the time for expert examination of each solution scales with $N^4$ for the polynomial method (for explicitly defining the background regions), and N for the spline method (for defining the EDF). Additionally, the quantitative criteria ($NAF$ and $\|\vec{a}_B\|_1^*$) that we introduce in this manuscript have been shown to sufficiently simplify the parameter selection process for users of any level of experience. We have included the following statement in the Conclusion section:

"Although the exact reduction in user time may be difficult to generalize due to high variability across different users, we reason the following approximation applies. Qualitatively, if $N$ values are considered for each free parameter in each method, then the amount of time for expert examination of each model solution scales up with $N^4$ for the polynomial method (due to 4 boundary points as free parameters) and $N$ for the smoothing splines method (due to 1 EDF parameter). Also importantly, the evaluation metrics, which we established in this manuscript, have been shown to sufficiently simplify the parameter selection process for users of any level of experience."

4. Abstract line 6: I suggest moving the word "remains" directly after "the question"

We moved "remains" after "of" since the preposition expresses the relationship between a part and a whole. The sentence now reads:

"Therefore, the question of how to develop an automated method for baseline correcting hundreds to thousands of ambient aerosol spectra remains given the variability in both environmental mixture composition and PTFE baselines."

5. Pg 2 line 8: "relatively" Pg 2 line 26: "A growing number of papers have been published in recent years" Pg 2 line 27: "One of the applications includes" Pg 4 line 8: "a priori. Therefore", Pg 14 line 13: omit comma

These have been fixed.

6. Figure 3: x-axis tick mark labels on left and right panels overlap. Figure 7 may be unecessary (especially given the large number of high-information figures that are certain to be large).

X-axis tick marks in Figure 3 have been adjusted and caption in Figure 3 has been modified to include:

"X-axis range from 4000 to 1500 cm$^{-1}$in both left and right panels."

Figure 7 has been moved to supplemental information section.